# Understanding implementation, adoption, and acceptability of the WHO package of essential noncommunicable (PEN) disease interventions in FIJI: Evidence for scale-up

**Bindu Patel** [1,2]*, **Devina Nand**[3], **Mohammed Alvis Zibran**[3], **Gade Waqa**[4], **Donald Wilson**[5,4], **Unise Vakaloloma**[4], **Rohina Joshi**[1,2], **Azeb Gebresilassie Tesema**[1,2], **Colleen Wilson**[3], **Stephen Jan**[1,2]

**1** The George Institute for Global Health, Australia, **2** School of Population Health, University of New South Wales, Sydney, Australia, **3** Ministry of Health & Medical Services, National Wellness Centre, Suva, Fiji, **4** Pacific Research Centre for the Prevention of Obesity and Non-Communicable Diseases, Fiji National University, Suva, Fiji, **5** Fiji National University, College of Medicine, Nursing & Health Sciences, Suva, Fiji

* bpatel@georgeinstitute.org.au

## Abstract

The Package of Essential Noncommunicable (PEN) disease interventions in response to the high prevalence of cardiovascular diseases and diabetes was implemented in Fiji commencing in 2012. This study aimed to understand implementation outcomes, and its contextual influences. Semi-structured interviews with health workers and patients across Fiji was conducted. Thematic analysis was mapped to the health system building blocks to understand PEN service delivery. The PEN program was well received by health workers formally trained. The frequency of use of PEN guidelines was influenced at the individual level by motivation, capability and capacity as well as external factors outside of the health workers' control. The key challenges to routine use were lack of essential medicines and equipment for CVD risk screening and management, shortage of health workers, high turnover of staff, limited formal training, and no designated focal person. However, at the country level, the PEN program improved the quality of care by providing patients with regular follow-up visits depending on their CVD risk levels. The patients found care to be comprehensive when they were also seen by dieticians and physiotherapists. In most instances, the barrier to access to care were cost and distance of travel and non-availability of essential medicines. To improve use of PEN program requires strengthening health system components: 1) need for efficient supply chain system for medicines and equipment, 2) improving healthcare workforce retention, 3) establishing accountability mechanisms embedded within the health centres, 4) health information system to track patient level data and 5) multi-level governance structures across the health system.

## Background

The Pacific Island country, Republic of Fiji (Fiji), is experiencing a rapidly deteriorating noncommunicable diseases (NCDs) crisis responsible for 84% of all deaths (50% of premature mortality)

**Data availability statement:** All relevant data contributing to the results are within the article and its Supporting Information files.

**Funding:** This work was supported by the National Medical Research Council Program Grant (#1149987 to SJ). The funders had no role in the study design, data collection and analysis, decision to publish, or preparation of the manuscript.

**Competing interests:** The authors have declared that no competing interests exist.

[1]. These deaths are mostly attributable to cardiovascular diseases (CVD) (34%), diabetes mellitus (22%), cancers (9%) and chronic respiratory diseases (5%) [2,3]. In response to the NCDs epidemic, the Fiji Ministry of Health and Medical Services (MHMS) supported by the World Health Organization (WHO) commenced implementation of an integrated package of essential noncommunicable disease interventions (PEN program) for those at high risk of CVD and diabetes.

The PEN program is a set of low-cost evidence-based interventions to address NCDs through primary care settings with limited healthcare resources. It includes essential technologies (diagnostics and equipment) and medicines to enable the use of several comprehensive packages of clinical services (through simple protocols) for early screening, diagnosis, and treatment for CVD, hypertension, diabetes, stroke, cancer, and asthma that include lifestyle counselling and health education (i.e. tobacco interventions, diet counselling, promotion of physical activity, avoidance of harmful use of alcohol), and regular follow-up plan [4]. Through this model, clinical services can be delivered to an acceptable quality of care by physicians and non-physician healthcare workers in low- and middle-income country (LMIC) settings [5].

In Fiji, the PEN program was integrated with evidence-based guidelines for diabetes management (now referred to as Cardiovascular Risk Assessment and Management (CRAM) guidelines) adapted for Fiji [6]. With strong political commitment to addressing NCDs in Fiji, PEN program was scaled up to become a national program. Implementation of two PEN protocols (protocols 1 and 2) commenced in all secondary level departments at the health centers and sub-divisional hospitals called Special Outpatient Departments (SOPDs) aimed at preventing CVD through the management of hypertension and diabetes (further details in *Methods* section on Fiji health system) [7].

The rationale for the PEN program at the SOPD level was to target high risk populations for prevention and treatment through early referral. The PEN program was piloted in Suva in 2012 and was scaled-up across all SOPD clinics in Fiji by 2019. The evolution of the implementation of the PEN program was governed from 2015 by a multidisciplinary National PEN Steering Committee and a National PEN Core Working Group (CWG) that was formed under the direction of the Head of Wellness at the MHMS. In March 2023, the PEN (CRAM) guidelines for Fiji was updated with a new Oceania chart for CVD risk reducing age for using PEN risk charts for those 30 years and over [8,6].

In LMICs, there have been several pilot implementation of the PEN program addressing CVD and diabetes yet findings on effectiveness and implementation process outcomes are mixed [8]. For example, in Bhutan the program was found to reduce CVD risk scores, smoking and alcohol use, hypertension and improved adherence to hypertensive medications. Similar findings were reported in Myanmar including reduced blood sugar levels. However, both studies had short follow-up periods between 3-18 months [9,10]. Studies with longer than 2 years follow-up found no difference in pre- and post- PEN protocol use for CVD and diabetes risk factors (Gaza/Palestine) [11]. There were critical challenges to the health system readiness in most of the studies with shortage of human resources, essential medicines and equipment at the primary care facilities [12]. In addition, a key issue in implementation were miscalculations CVD risk scores, lack of adherence to the PEN protocols and lack of patient follow-up [8]. These findings provided Fiji an impetus to evaluate implementation processes, health system readiness and patient understanding of CVD risk [13].

The MHMS is planning to expand the PEN program to primary care levels for early detection and prevention. To ensure this expansion is successfully implemented it is important to understand how it is has been managed thus far. The aim of this process evaluation of the PEN program in Fiji seeks to unpack the underlying mechanisms of implementation of PEN by addressing questions of how, why, by and for whom and under what circumstances the PEN program was used or not used and how it was accepted by patients.

## Methodology

### Study design

The evaluation drew on the RE-AIM framework logic model (S1 Appendix) to assist in the planning, design and conduct of the evaluation [14,15]. The process indicators for data collection that were considered were based on participant **R**each, **E**ffectiveness of the PEN program, **A**doption by healthcare workers, **I**mplementation fidelity of the service delivery, and **M**aintenance of the PEN program over time [16].

A case study methodology was adopted with the SOPDs serving as the unit of analysis enabling exploration of implementation processes from multiple sources and contexts [17]. We purposively selected SOPDs to achieve maximum variation in staff numbers at each SOPD, patient profile and community demographic through consultation with the co-authors and Steering Committee that was formed for the evaluation of doctors, nurses, dieticians, implementers, and researchers in Fiji. We purposively sampled participants (healthcare workers and patients) from the SOPDs until we reached thematic saturation. The interviews took place at the SOPD using a flexible interview guide (S2 Appendix) that allowed exploration of emergent themes.

### Study setting

The Fiji Islands are spread over four regional Divisions (Central, Western, Northern and Eastern) and divided into 21 sub-divisions with a population of ~900,000. About 57% of the people are iTaukei followed by 37.5% Indo-Fijians, 1.2% Rotuman, and 4.5% other ethnicities [18,19]. The health system is based on a three-tier model that provides integrated health services at primary, secondary and tertiary levels. There are three divisional hospitals, 21 sub-divisional hospitals, 84 health centers, and 98 nursing stations (Table 1) [20]. The MHMS is responsible for providing accessible, affordable (free or subsidised healthcare), efficient, and high-quality healthcare to all people across Fiji [21].

### The cases resided in the four Divisions and six Sub-divisions (Fig 1).

Study objectives

1. To identify and understand the processes that influenced adoption of the PEN program by healthcare workers, and key contextual factors that influenced PEN use as routine clinical practice [22].

Table 1. MHMS Health System structure [20].

| Health Facility | Central Division | Western Division | Northern Division | Eastern Division | Total |
|---|---|---|---|---|---|
| Specialized hospitals/National Referral | 2 | 0 | 0 | 0 | 2 |
| Divisional Hospitals | 1 | 1 | 1 | 0 | 3 |
| Sub-divisional Hospital [level 1] | 0 | 3 | 1 | 0 | 4 |
| Sub-divisional Hospital [level 2] | 5 | 3 | 2 | 5 | 15 |
| Health Centre [level A] | 7 | 4 | 1 | 0 | 12 |
| Health Centre [level B] | 2 | 4 | 3 | 1 | 10 |
| Health Centre [level C] | 12 | 20 | 16 | 14 | 62 |
| Nursing Stations | 21 | 25 | 21 | 31 | 98 |
| **Total** | **50** | **60** | **45** | **51** | **206** |

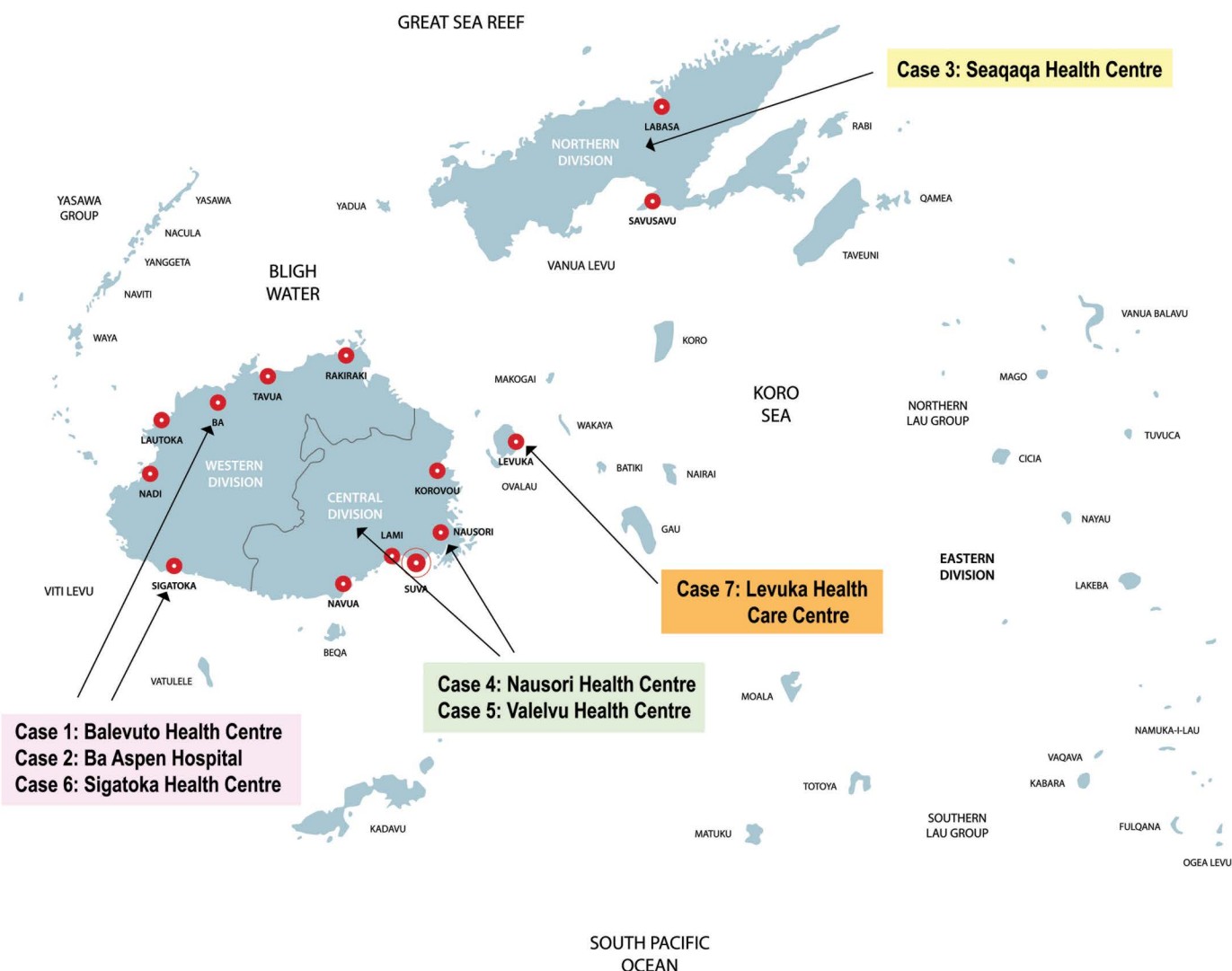

**Fig 1. Participating Special Outpatient Departments (cases) at health centres and hospital in FIJI (credit: iStock.com/Royalty Free).**

2. To assess barriers and facilitators to implementation fidelity of the PEN program, the degree to which the multifaceted PEN program was implemented as intended, a critical initial link in the implementation strategy to PEN use (service delivery) [23].

3. Explore patient access and acceptability of the PEN program [24].

## Study participants

A total of 27 healthcare workers (doctors, nurses, dieticians, and physiotherapists) and 23 patients across the four Divisions (six sub-divisions) were invited to participate in the study (Table 2 and Table 3). The data collection took place between July 2, 2022, and April, 30, 2023 at the health centres. Participants were provided participant information sheet and consent form with ample time to review and ask questions. Consent was both verbal and written. All participation was voluntary.

**Table 2. Socio-demographic profile of healthcare workers.**

| Socio-demographic variables | Frequency (n = 27) |
|---|---|
| *Age of participants (years)* | |
| ≤29 | 11 |
| 30-39 | 13 |
| 40-49 | 3 |
| *Mean age (SD): 32 years (±5)* | |
| *Sex* | |
| Male | 6 |
| Female | 21 |
| *Profession* | |
| Doctor | 9 |
| Staff Nurse | 9 |
| Dietician | 6 |
| Physiotherapist | 3 |
| *Length of years in health workforce* | |
| ≤ 5 years | 9 |
| 6–10 years | 11 |
| 11–15 years | 5 |
| 16–20 years | 1 |
| > 20 years | 1 |
| *Heard about PEN during undergraduate/post-graduate training* | |
| Yes | 5 |
| No | 22 |
| *Received formal training on PEN* | |
| Yes | 14 |
| No | 13 |

## Data collection

The healthcare worker's interview framework allowed to generate evidence on staff understanding and their engagement with the PEN program, barriers and facilitators of becoming routine clinical practice, and how organisational level context influenced uptake of the PEN program across the different divisions. The interviews with patients explored quality and access to care, awareness of the PEN program/CVD risk and their engagement with treatment recommendations. This allowed us to identify barriers and facilitators of the PEN program service delivery at the consumer level. We worked to ensure ethnicity was representative of the population based on sub-division and SOPD demographics. We used maximum variation sampling methods also referred to as maximum diversity sampling with respect to age, gender, and ethnicity.

## Data analysis

The interview data had three stages of analysis assisted by NVivo 20 (QSR International Melb. Vic), a data management tool. The first stage was familiarisation with initial transcripts from two researchers (co-authors, UV and BP). In the second stage, two researchers (BP and UV) became immersed in the data by listening to audio recordings, reading transcripts, and documenting emergent themes. Finally, the themes were deductively and inductively coded by authors BP and AGT based on the interview framework and then synthesised using

**Table 3. Socio-demographic profile of patients interviewed.**

| Socio-demographic variables | Frequency (n = 23) |
|---|---|
| *Age of participants (years)* | |
| 21–40 | 2 |
| 41–50 | 4 |
| 51–60 | 11 |
| 61–70 | 4 |
| 71–80 | 2 |
| *Mean age (SD): 55.8 years (± 11.2)* | |
| *Sex* | |
| Male | 11 |
| Female | 12 |
| *Marital Status* | |
| Single | 1 |
| De facto | 1 |
| Married | 21 |
| *Highest academic level* | |
| Year 10 or below | 8 |
| Year 11–12 | 14 |
| University undergraduate degree | 1 |
| *Employment status* | |
| Unemployed | 13 |
| Part-time/Casual Work | 1 |
| Full time work | 1 |
| Permanent | 1 |
| *Accessibility* | |
| Phone | 21 |
| Car | 13 |
| Computer | 5 |
| Internet | 9 |
| *Accommodation type* | |
| Flat owned by family | 1 |
| Sister's place | 1 |
| Leasing & renting | 2 |
| Own home/Flat | 19 |
| *Ethnicity* | |
| Fijian (iTaukei and Rotuman) | 8 |
| Indo-Fijian | 13 |
| Others | 2 |

Mikkelsen-Lopez et al's adapted WHO health system building blocks framework [25] (Fig 2, S3 Appendix) linking governance to medicines and technology, information, health-care workforce and ultimately service delivery with the overarching goal of equitable access to care for the "people" to improve CVD and diabetes outcomes. Phrases or full sentences that most accurately expressed the categories under each theme were then identified and presented as quotes in the results section.

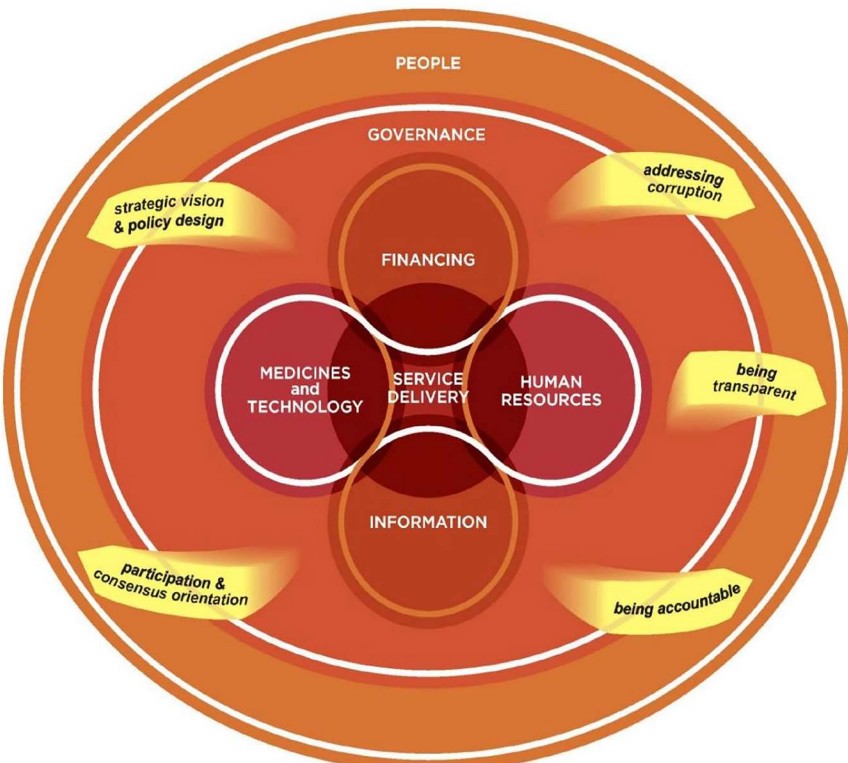

**Fig 2. A systems thinking perspective to assessing health system performance.Credit: Mikkelsen-Lopez, I., Wyss, K. & de Savigny, D. An approach to addressing governance from a health system framework perspective. BMC Int Health Hum Rights 11, 13 (2011).** https://doi.org/10.1186/1472-698X-11-13.

## Ethical considerations

Ethical approval was obtained from the Fiji Ministry of Health and Health Services (#10/20222) and the University of New South Wales (UNSW) human ethics research committees (#HC220213).

## Results

Our findings are organised into three main domains and respective overarching themes, as outlined in Table 4. Additionally, key enablers and barriers for each case are provided in S4 Appendix.

### Domain 1: Adoption of PEN program by healthcare workers

**1.1. PEN protocol use.** The use of one or more components of PEN protocols was influenced by factors that can be readily mapped to the health system building blocks.

Protocol 1 Use (Management of CVD through integrated hypertension and diabetes care): Protocol 1 required screening for and calculating CVD risk, and then prescribing treatment, as appropriate. The delegation of responsibilities according to the interviews in most instances was that the nurses would screen and create colour-coded patient folders based on the colour of their risk based on PEN CVD risk chart. The doctors calculated CVD risk that were dependent on many factors, including time and formal PEN training.

**Table 4. Domains and themes for implementation, adoption, and acceptability of PEN.**

| Domains | Themes |
|---|---|
| 1.Adoption of the PEN program by healthcare workers | PEN protocol use |
| | PEN protocol use in community-level outreach programs |
| | Perceived impact of PEN |
| 2.Health system components facilitators and barriers of PEN service delivery | Governance and Leadership |
| | Medicine and Technology |
| | Health workforce |
| | Information Systems |
| 3.Patient's healthcare access and acceptability | Access to care |
| | Acceptability of PEN |

"… and then I did his[patient] physical examination, history taking, and I gave him [xx] some blood forms and I already started him on medications. Then he will come back to us for reviews. Once he gets his bloods and everything, then we will fill his PEN model form". – 01-003, doctor (Western division)

Participants highlighted the importance of the PEN tool in guiding healthcare workers to provide treatment for individuals without established CVD or diabetes for prevention, as well as ensuring continuous follow-up for all patients, and medication refills for those at high risk. A nurse from Northern division (03–003) shared '*… We also do blood test here, validations. So, looking at the blood test and the colour coding then we add the statins."*

PEN Protocol 2 Use (Health education and counselling on healthy behaviours): Dieticians and physiotherapists provided health education, counselling on healthy behaviour and pre-scribed appropriate lifestyle modifications according to their risk levels. A physiotherapist from Western division (06–004) shared, '…we do counselling… they also talk and it's not just a one person, it's a talk. And that's where we also fill in this form [PEN] so that we can gauge their lifestyle'. Another physiotherapist from Western division (02–002) added,

"All these SOPD cases they are referred by the doctors to us. We usually prescribe them with some exercises…totally different from what we give a patient with a green folder and what we give to a brick red folder."

In addition, dieticians supported health facilities by providing heart-healthy diet education and encouraged healthy lifestyle behaviours. One dietitian (04–006) from the Central division highlighted how PEN strengthened their approach through the evidence-based guidelines to addressing dietary issues moving the risk levels from red to green.

"By looking at the PEN model approach, my aim, as a dietician, if there's a patient who comes in and we put the colour code and then, for example, that patient was sitting on red, my aim is to get that red colour coding to green by marketing my nutrition therapies…I had some change…It worked. Some came from red, brick red, to green."

Furthermore, participants highlighted that the use of motivational interviewing technique with the patients influenced behaviour change. A dietician (04–006) indicated,

"There was this motivational interviewing concept that we had to go for, we undergo this training. So, I just told them, "Okay, you've been sitting on overweight, now you have gone

up to obese, can you do something about it?" So, I'll let them identify themselves what they can do for them to achieve their normal weight."

**1.2. PEN protocol use in community-level outreach programs.** The PEN protocol was used at the community level as part of outreach programs depending on staff availability. A healthcare worker from the remote Eastern division (07 – 001) shared '*I requested community health workers to ask the SOPD patients in their villages if they were interested that I come and hold a session and it happened. It worked in one village and then, I moved into another village…*'. A healthcare worker from Northern division (03–005) highlighted,

"When we do community outreaches, we are able to detect it quite earlier. Just recently I detected one high BP and this gang here [at the health centre] missed it…So we do community outreaches. We have one nurse allocated that comes with us."

However, such outreach visit varied between divisions. As participants highlighted, there were more outreach visits in the Eastern division than other areas due to factors including patient cost and distance of travel to the health centre.

"…'my transport returns back 12:00 midday', I [the patient] have to make sure I make them seen by an MO [medical officer] by 12:00 midday. They have to be seen. [The HCWs] are going out to clinics now because they [the patients] have got the habit of outreach programs. I have been doing outreach for two years, 2021 and half of 2022 till April. So till April 2021, 2022 April, we were only doing outreach. There were no clinics done in the hospital. So they know the nurse will come with the doctor, with the dietician, with the physio and their medications. The outreach, we go to the village, so they know that the nurse will come to their village." - 07-002, nurse (Eastern division)

**1.3. Perceived impact of use of PEN protocols.** Most participants believed that the PEN program has assisted in continuous care of patients referred to the SOPD. A doctor from the Northern division (03–005) shared that the PEN program helped to standardize in CVD assessment and management '*I think it's a good idea that we have the PEN model because everyone is on the same page…*'. Another doctor from Western division (01–003) added that,

"I think it [PEN] has worked. Before PEN, …they[patients] just get managed, and they don't get followed up and they don't really get classified in terms of the risk and they don't get the management based on their risk factors. But after the introduction of the PEN, now we know whether patients are at low risk, medium and high risk and what we need to focus on at which risk we need to give them. Which treatment or which management where we must go from."

However, a few participants had different perspectives, as they felt that while training doctors has been useful, it has not yet translated to better patient outcomes, and they did not believe in the effectiveness of the PEN program. A doctor from Western division (02–001) shared '*Truthfully? No, [has not made a patient impact]. I mean, we're doing our part. We're trying to improve our assessment of patients, our treatment for patients but has this lowered our numbers? No*'.

## Domain 2: Health system components facilitators and barriers of PEN implementation

Below we outline the health system enablers and barriers to use of PEN. These factors are essential for positively influencing PEN program service delivery and its impact on the patient population.

**2.1. Governance and Leadership.** The PEN program governance structure (core working group) supported implementation through delivering training to healthcare workers, supply of equipment and medications at divisional and sub-divisional levels and provision of regular clinical audits. The CWG's leadership and advocacy for PEN program has been instrumental in integrating PEN as a core service delivery at the SOPD levels throughout Fiji. However, there were key organisational barriers to the PEN service delivery. One of the key challenges hindering regular use of PEN program was lack of leadership and operational structures for accountability and clear lines of communication and feedback loop to health centres for PEN related queries. To assess and strengthen compliance, PEN audits were performed by WHO PEN officers. However, the focus was on assessing the proper use of the evidence-based risk tools rather than facility readiness according to interviews. Feedback on findings from the clinical audit, and process for improvement for the health facilities was limited. The healthcare workers were uncertain how the information is fed back to the MHMS and then back to the health centres to ensure they have proper resources necessary to delivery PEN. *'Normally when they [WHO PEN staff] come, they don't focus on equipment. They mainly focus on the folder'* stated a nurse from Western division *(*01–002*).*

Leadership and coordination processes related to PEN activities faced many challenges at a health facility level. As discussed by the participants, most SOPD didn't have a designated PEN focal persons to ensure effective oversight to PEN protocol implementation. The leadership on PEN program across most Divisions was voluntarily undertaken by trained staff within the facilities, and this was usually interrupted with staff turnover due to transfer or emigration. Participants also highlighted the lack of a clear accountability mechanism to ensure compliance, such as performance rewards and the imposition of sanctions, as another possible reason. A doctor from Western division (07– 003),

> "…We[xx] get transferred somewhere else and the new staff comes in. And then the continuity of PEN Model within the facility depends on the staffing. Those that have been trained if they have gone and the ones that are running SOPD now if they haven't been trained, there's going to be a gap in terms of maintaining PEN."

**2.2. Medicine, Equipment and Resources.** The availability and adequacy of medicines and equipment as well as the functionality of the equipment, varied across the divisions and health centres. Supply for essential medicines to health centres were meant to occur quarterly from The Fiji Pharmaceutical & Biomedical Services (FPBS) via email to the MHMS. However, supply was often unreliable, depending on stocks at the national level. A nurse from the rural Western division highlighted (01–002), *'It depends on their supply. If FPBS have a lot of Metformin, they'll supply us more. If they have less, then they have to minimise and use it in other health centres'.* Further, the supply of medications often depended on where the health centre was located, as there were minimal issues with medications supply in urban health centres. A SOPD doctor from this division (04–003) indicated that,

> "…but most days, the medications are available. Most of the medication's available. With the recent round that I did, I think in most of the facilities we had about 70 to 80% of that medical supply was available." - 04-003, doctor (Central division)

A significant finding from this evaluation highlighted that the availability of medicines being key to sub-optimal patient care. A nurse from rural Western division (01–002) shared *'Balevotu, is based on farming background and patient don't normally go and buy, only the ones that are a bit educated, they can afford, they go and those who are concerned about their health'.*

Several participants indicated the challenges with limitation of essential PEN equipment and resources, such as glucose strips, blood pressure machines, and PEN risk charts thus impacting use of the PEN program.

> "Probably more equipment. I mean if, I think the Ministry of Health is involved in it so it might work to MHMS and then we can get more equipment and more trainings for staff. And the most important, our SOPD needs that PEN chart…. I think there's one with the nurse, but we need more." - 06-001, doctor ((Western division)

In addition, other facility resources posed significant barriers to day-to-day care of patients. This included not having any space for counselling, and transportation for outreach program for nurses, dieticians and physiotherapist. As indicated by the healthcare workers, it was important for health centres to ensure transport availability to get to the communities for outreach clinics. In the Northern division, the nursing manager developed a system for the limited transport allowing drop off and pick up of different healthcare workers in the different villages throughout the day. In addition, in the Eastern division, SOPD clinic outreaches were scheduled on different days to ensure adequate participation and transport; *'we did not have transport so we have to liaise with police, with other government workers if they can give their transport to us for us to pick and drop…' (07-002, nurse).* Despite the transportation challenges, participants emphasised the importance of the outreach program to ensure access to care for community members.

> "Now trying to send health workers every day outside so that they[patients] are happy that our services are reaching them because some of them are in the geographical location where there is no transport, and they can't reach our health facility. Although we have only one transport, we make use of it to reach them to reach patients through community visits." -03-002, nurse (Northern division)

**2.3. Healthcare workforce.** Motivation and capability: A motivated and capable experienced allied healthcare worker was instrumental to the implementation of PEN protocol 2 in the Northern division. The physiotherapist worked across many communities and health centres developing strategies on how best to implement protocol 2 by risk levels (protocol 1). The physiotherapist developed a template that combined CVD risk stratification with physical profiles to prescribe exercise;

> "I was looking after the PEN cases there…the counselling and assessing their physical activity level. We have come up with a template, we've made a template and I believe that's only for Northern. You won't see that template elsewhere." –03-001, physiotherapist (Northern division)

Staff shortage/turnover: '*…it [PEN] was not done here for more than a year now. One doctor and one nurse were trained but that medical officer who was trained took the post of senior medical officer so he's busy with that…'(03-002, nurse, Northern division).* Another participant from Western division (06 – 001) alluded to the staff shortage,

> "We have only one physio who does GOPD [general outpatient department], inpatient and SOPD. So sometimes there's a lot of beds, so the physio cannot be available for advice to patients for physical activities and the other one is the counsellor. We don't have the counsellors available. So, we do whatever counselling we do by ourselves or the SOPD nurse, that's it."

Staff training: Many healthcare workers were not trained on the PEN program. Although, some healthcare workers learnt by observation and interaction with the trained staff to use the intervention to some extent, a high turnover of trained staff prevented the use of the PEN intervention to its full extent. Some were motivated and learned through available resources, '*whatever resources that I had in the SOPD, I went through it to learn…*' stated a doctor (03-004, Northern division). Time constraints of doctors and nurses also affected the quality of care. '*Here in Valelevu there is a lot of patients being booked. I feel that I don't have enough time for all these patients*' (05-003, doctor, Central division).

**2.4. Information Systems.**  According to one interview, health centres recently began the collection of NCD data that is sent to the divisional and national levels. However, it is not clear how data generated from PEN is integrated into the NCD database. More so, the data being collected is cross-sectional and there is currently no system to monitor patients' progress over time.

> "Part of the data is not just the number of cases attendance, we are trying to trace up our defaulters, and also how many get transferred in from other facilities to our facilities. There's a whole template, the Divisional Medical Officer's template for reporting. That's what we are trying to collect for the whole subdivision. All that goes up to division and it gets reported to national." -04-003, doctor (Central division)

Further, participants signalled that there is yet to be a systematic and unified process for PEN data collection and management across divisions and sub-divisions. A reason for this is because NCD data collection is paper-based and challenging to integrate into one system across divisions for monitoring and evaluating key indicators for action. '*…but most of them[xx] are still using the paper copy so it's difficult… it'll be easier to get electronic copy*' (04-003, doctor, Central division).

A key enabler PEN program was that it improved the quality of continuous care by using PEN to systematically categorise the patient folder based on colour (green, yellow, orange and red) of their CVD risk levels. This assisted doctors and nurses to know who needed more frequent follow-up and medication re-fills. Although, it is currently difficult to monitor for improvement on biological risk factors for hypertension and diabetes in a longitudinal manner given the lack of established electronic data management system, participants agreed that PEN categorisation helped them to advise patients according to their risk level.

> "PEN, it helps to identify the patients, categorise the patients and manage them accordingly. When you categorise, the patients based on their classification based on their risk factors, we are able to treat them and manage them better than we might when we just see them without any system and at the time try to reduce the mortality and morbidity associated with NCDs." -01-003, doctor (Western division)

## Domain 3: Patient's healthcare access and acceptability

**3.1. Patient experience.**  **Access to care:** Access to care is a concept that involves both supply (health service) and demand-side (consumer) considerations. These dimensions of access were often interacting, '*for instance, we would have to pay a fare if we had to visit the hospital for a checkup. In contrast, we are not required to pay for our travel when the health team comes to our village to perform our [SOPD] checkup*' (07-01-01, patient). Our study identified several factors affecting patients' access to care, including their perceptions of healthcare providers, geographic location and availability of community outreach, and the cost of travel to attend regular SOPD visits.

**Patient perceptions of healthcare providers:** Patients' experiences with healthcare providers were mixed where some found the staff to be thorough and caring, and others not being provided sufficient explanation to make shared decisions on their health needs. A participant from Western division (01-03-03, patient) shared '*doctors and nurses are very good, they proper check. Whatever pains, we tell them, and they check…*'.

However, some perceived they were not getting comprehensive care on their CVD risk; *'no, I mean they do not discuss anything they just read our report and then whatever our blood sugar and blood pressure level is…They will just tell you and they will just prescribe the medicine' (06-01-01, patient).*

**Appropriateness of care:** PEN program was dependent on multidisciplinary teams with health education and counselling key to health promotion for prevention of NCDs. Dietary advice by dieticians was significant in influencing heart-healthy diet especially using motivational interviewing technique. A patient (03-01-01, Northern division) shared '*… before I was eating plenty meat, now I'm eating more vegetables…. I take no sugar with my tea, and I don't take lollies …*'. Another patient (04–01-01) shared similar positive experience '*the dietitian is very important and due to that, when I came for next checking, my sugar went from ten to eight. I cut down. I was having little pineapples and things like and that was because of the dietitian*'.

Access to care increased with the PEN program as a result of regular follow up visits that allowed to monitor change in behaviour, medication compliance and CVD risk status.

"You know we're Fijians yes, we believe in herbal medicines, yes. But doctor told it's better you take your pills. So, I don't want to miss it, yeah, because I was working so hard, I had to come, go to SOPD… so, I have my check in the clinic after three months." -04-03-03, patient (Central division)

**Community outreach for service delivery:** Participants considered healthcare workers' community outreaches to be beneficial to prevention of disease and health promotion. Early detection and prompt management of asymptomatic diabetes and other NCDs. A participant from Eastern division (07–01-01) described '*…in fact the doctors visited our village on Tuesday, and we got our checkup done. Usually, each village has its own certain days for clinic*'.

"Last week the health team visited our community along with the Minister of Health. The services they brought to the community included tooth extraction, pressure taken and others. We are very happy about their visitation as we don't need to go to the health centres for our check-ups as some cannot afford to do so." -03-04-04, patient (Northern division)

In rural and remote regions, bus schedules were often limited to once a day. Although there were some variations between division, as indicated in Domain 2, various health centres provided transportation to reach their respective communities for outreach. For example, in the Eastern division, SOPD outreaches were scheduled on different days to ensure adequate participation. A dietitian from Northern division (03–005) highlighted that '*We dispense medication, but we can't basically follow them up because the reason we go to them is because they can't come to the [health centre]. Mostly because of financial problems or demographic problems*'.

In addition to transportation, the participants denoted that patients in most places were frustrated with the long wait times at the health centres especially when public transport was limited, '*waiting time needs to be considered as some of us come early in the morning and wait for 3 to 4 hours to get checked. So, more doctors are needed*' (03-01-01, patient Northern division). Another participant (07–02-02) also added,

"I think the services are delayed at times. The waiting periods are long, considering the nurses start at any time that's convenient for them. Usually, the opening time should be eight, but then at times we don't start at the exact time, but we wait longer."

**3.2. Acceptability of PEN program.**  Although patients visit SOPDs, most participants were not aware of their CVD risk or the PEN program. Some were aware of the colour coded folders but not sure what it meant. A patient from Central division (05–01-01) highlighted this '*No [I don't know what my CVD risk] …I thought when they brought the file the color had no meaning*'.

## Discussion

The PEN program was implemented at the national level in all SOPDs across the primary care health centres with variable levels of use. The use was influenced by the multiple interacting and dynamic health system components to deliver PEN program efficiently and effectively. The routine use was hindered by several essential PEN components, including limitations in the availability of essential medicines, basic diagnostic equipment, adequate space for counselling, and a sufficient healthcare workforce. Additionally, challenges such as resources for outreach programs to reach patients in the communities, the absence of an information system to monitor patient-level data over time, lack of a focal person for PEN related activities, feedback loops and accountability at the health centres further exacerbated the barriers to use. Leadership continuity was disrupted as a result of staff turnover at health centres that prevented adoption of PEN as standard care, and in turn limiting access and acceptability of PEN program by the patients. In addition, use of evidence-based guidelines as intended was sub-optimal, with poor recording of CVD risk. A key reason was because of the high turnover of staff that were trained, and lack of training of new staff due to resource constraints.

Furthermore, participants noted that the community outreach initiatives significantly increase access to care for individuals who would otherwise be unable to travel to health centres due to multiple factors. Socio-economic demographics play a crucial role in shaping access to care. Many Fijian patients face challenges such as distance, cost, work schedule along with limited bus frequency particularly in rural and remote regions. To address these barriers, it is important for health centres to ensure availability of transport for healthcare workers, enabling them to conduct outreach programs.

The findings indicate that there is a need for strengthening and building a resilient health system for PEN program to be delivered effectively. Fiji's national health strategy is to reduce the burden of hypertension and diabetes, reduce healthcare costs and human suffering through increased population coverage to prevent, screen, and manage risk factors for CVD and diabetes. In the Global Action Plan (2013-2020), WHO included a target of >80% affordable availability of essential medicines and basic technologies required to treat major NCDs in public and private facilities, and core set of seven diagnostic tests that need to be available (blood glucose, blood cholesterol, urine albumin, urine glucose, urine ketones, serum creatinine, and serum troponin) [26]. To strengthen primary health care [PHC] systems, and monitor core health system indicators, there is a need to incorporate a 'PHC measurement conceptual framework and indicators' to the program logic that can be tailored for Fiji, and monitored and evaluated continuously [27]. In addition, the goal of the intervention is improving the access to integrated care for community members for prevention and control of hypertension and diabetes at the PHC facilities in Fiji through the PEN protocols 1 and 2 by a multidisciplinary team of the doctors, nurses, dieticians and physiotherapists. This also requires a continuous monitoring and evaluation framework for facility and patient level data for core NCD indicators to be applied to the PEN program to assess effectiveness [28].

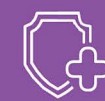

**Recommendations to adopt, sustain and scale up the Package of Essential Noncommunicable (PEN) disease program in FIJI**

**Strengthening and building resilient health systems for accessible, continuous and comprehensive PEN service delivery require the following:**

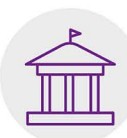

**Leadership and governance** are critical levers required for strengthening health systems. In addition, multisectoral (e.g. education, agriculture, communication, transport, finance) collaboration with sectors outside of health are required and a mechanism in place to enable evidence-based policies and strategies to be enacted across the sectors. This includes implementation plan, and governance structures across the different levels of the health system for clear accountability and feedback loop of PEN related activities and communication.

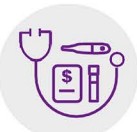

**Digital medicines and equipment** procurement and supply chain system is required at the national level for NCD commodities. This includes improved forecasting, tracking and communication between health facilities and FPBS for NCD medicines and basic equipment for screening. A system has been rolled out at the national level in Fiji at the end of December 2023 for medicines and equipment at all health facilities.

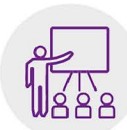

For PEN service delivery, **strengthening of training** and mentoring program through development of core competencies framework (standard of care) required for different healthcare cadres. Development and implementation of multi-method learning modes such as classroom, hands on practical training with supervision, regular e-learning videos, team huddles and embedding quality improvement component. This will ensure no gaps in training for new staff and continuous support for staff already trained.

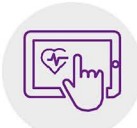

Implementation of a **digital information system** to record patient level data that include screening, detection, treatment and follow up visits. A key feature to the tool should have an automatic CVD risk calculation and treatment recommendations. This will allow to monitor and evaluate clinical audit data across the health center facilities. In addition, have a patient risk communication dashboard to assist health care cadres in providing risk specific treatment and lifestyle advice for patients to understand their CVD risk and rationale for treatment for prevention and management.

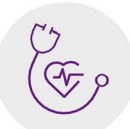

Strengthen resources for **continuous community outreach programs** to support nurses and CHWs. Our findings showed that socio-economic demographics of the patient population affect access to care, with many patients unable to reach health centers due to distance and cost.

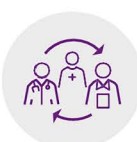

**Establishment of efficient and effective team-base care model with task sharing of screening to nurses and CHWs.** NCD prevention and management require continuous and comprehensive healthcare support, early detection through evidence-based screening, risk factor identification, self-management, lifestyle modification and regular healthcare support. There is substantial evidence that primary health systems have significant potential to improve healthcare outcomes with socially and culturally acceptable intervention methods such as PEN program through the community health workers and nurses.

**Fig 3. Recommendations to adopt, sustain and scale-up PEN program in Fiji.**

We have summarised our recommendations in Fig 3 based on our findings on how Fiji can effectively deliver PEN program that is sustainable. The limitation of this study is the lack of patient level data to triangulate the findings with the process evaluation. However, the strength of the process evaluation was that data was collected from all four divisions including

maritime region that provides a good basis for generalisation for the entire country on how the PEN program is being used and accessed by the communities.

## Conclusion

In Fiji, the strong political commitment from key government leaders and multi-lateral and donor agencies was critical to national level 'adoption' of the PEN program with key mechanism of adoption being the PEN CWG guiding the implementation team. While the PEN program has achieved some desirable outcomes, the study highlighted areas of the Fiji health system that requires further strengthening, including strengthening multi-level governance mechanism, health centres' operational and management structures, commodities procurement, digital information system to track patient level data over time, and health promotion and preventive care at the community levels. This requires clear ownership and accountability at the different health system levels and lines of communication across from MHMS implementation team and governing bodies to the health centres and communities. Further multi-sectoral and multidisciplinary partnership involving medicines, equipment, IT infrastructure, healthcare workforce, training, financing and governance will be required to ensure alignment of agenda and funding at the national level for improving health and wellbeing outcomes and work towards universal health coverage.

## Supporting information

**S1 Appendix:  Logic model.**
(DOCX)

**S2 Appendix:  Interview guides.**
(DOCX)

**S3 Appendix:  Mind map.**
(DOCX)

**S4 Appendix:  Evidence by cases.**
(DOCX)

## Acknowledgments

We would like to thank the evaluation Steering Committee members (Shrish Acharya, Swastika Chandra, Nemani Seru, Ledua Tamani, Filimone Raikanikoda, Alvina Deo, Lisi Finiasi, Ateca Kama, Momtaz Ahmed, Abdul Shah, Amos Zibran, Luke Nasedra, Pratima Singh, Shaileen Ahmed, Sarote Nakaora, Violet Vereivau, Sainimere Vulibeci, Josaia Tiko, Tomo Kanda, Anna Palagyi, Jacqui Webster, Mereani Tuibua, Salome Maiwaikatakata, Maria Maivalenisau, Ana Kalokalo, Merelita Taiserua) who were critical to the design of the evaluation, collection of data and discussion and input on the results. We would like to provide a special thank Rohin Latchmi, the WHO/MHMS PEN Officer, Senior Divisional Medical Officers, Senior Divisional Nursing Managers and all the health professionals and patients that openly shared their perspectives, knowledge and critical insights to the researchers. These findings would not have been possible without these individuals. In addition, we would like to thank Whenayon Simeon Ajisegiri for his contributions to assisting with validating the thematic analysis.

## Author contributions

**Conceptualization:** Bindu Patel, Devina Nand, Stephen Jan.

**Data curation:** Bindu Patel, Unise Vakaloloma, Rohina Joshi, Azeb Gebresilassie Tesema.

**Formal analysis:** Bindu Patel, Azeb Gebresilassie Tesema.

**Funding acquisition:** Stephen Jan.

**Investigation:** Bindu Patel, Unise Vakaloloma, Rohina Joshi.

**Methodology:** Bindu Patel, Mohammed Alvis Zibran, Donald Wilson, Rohina Joshi, Azeb Gebresilassie Tesema, Stephen Jan.

**Project administration:** Bindu Patel, Mohammed Alvis Zibran, Unise Vakaloloma.

**Resources:** Gade Waqa.

**Supervision:** Bindu Patel, Devina Nand, Gade Waqa, Donald Wilson, Colleen Wilson, Stephen Jan.

**Validation:** Bindu Patel, Devina Nand, Mohammed Alvis Zibran, Unise Vakaloloma, Rohina Joshi, Azeb Gebresilassie Tesema, Colleen Wilson.

**Visualization:** Bindu Patel, Azeb Gebresilassie Tesema.

**Writing – original draft:** Bindu Patel, Azeb Gebresilassie Tesema.

**Writing – review & editing:** Bindu Patel, Devina Nand, Mohammed Alvis Zibran, Gade Waqa, Donald Wilson, Unise Vakaloloma, Rohina Joshi, Azeb Gebresilassie Tesema, Colleen Wilson, Stephen Jan.

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
