## [Decision Letter · Decision Letter 0]

7 Feb 2025

Understanding implementation, adoption, and acceptability of the WHO Package of Essential Noncommunicable (PEN) disease interventions in FIJI: evidence for scale-up

PGPH-D-24-02590

Dear Dr. Bindu Patel,

We are pleased to inform you that your manuscript 'Understanding implementation, adoption, and acceptability of the WHO Package of Essential Noncommunicable (PEN) disease interventions in FIJI: evidence for scale-up' has been provisionally accepted for publication in PLOS Global Public Health.

Best regards,

Man Thi Hue Vo, MD, PhD

Academic Editor

Reviewer Comments (if any, and for reference):

Reviewer's Responses to Questions

**Comments to the Author**

1. Does this manuscript meet PLOS Global Public Health’s publication criteria ? Is the manuscript technically sound, and do the data support the conclusions? The manuscript must describe methodologically and ethically rigorous research with conclusions that are appropriately drawn based on the data presented.

Reviewer #1: Yes

Reviewer #2: Yes

Reviewer #3: Yes

2. Has the statistical analysis been performed appropriately and rigorously?

Reviewer #1: Yes

Reviewer #2: Yes

Reviewer #3: I don't know

3. Have the authors made all data underlying the findings in their manuscript fully available (please refer to the Data Availability Statement at the start of the manuscript PDF file)?

Reviewer #1: Yes

Reviewer #2: Yes

Reviewer #3: Yes

4. Is the manuscript presented in an intelligible fashion and written in standard English?

Reviewer #1: Yes

Reviewer #2: Yes

Reviewer #3: Yes

5. Review Comments to the Author

Reviewer #1: Dear Authors,

After careful consideration of the text with three figures and the three attached graphic and table files, I find the manuscript to be very interesting, well-written and, above all, of a high application value for improving the effectiveness of Fiji’s health system in efficient delivery of the PEN program in the Fiji healthcare system.

The content of the manuscript was based on a solid theoretical foundation, the study was conducted on the basis of a correctly applied methodology, the analysis of the results was clearly presented.

The results obtained were discussed in relation to indicated three main objectives, and on this basis conclusions were developed in a concise and clear form.

The manuscript meets the requirements of the Journal, and in my opinion, is ready for publication in the PLOS Global Public Health.

Kind regards,

Reviewer

Reviewer #2: The manuscript as reviewed meets PLOS Global Public Health publication requirements. The authors presented the study methods, results, discussions and conclusions in a clear and effective manner. No revision necessary and this manuscript is accepted and recommended for publication.

Reviewer #3: The paper is predominantly a descriptive/qualitative study. Although Tables describe the participants profile in terms of numbers of health care professionals and patients participating in the study, the study does not categorise any of the observations in terms of the parameters of the study participants.

Another drawback is that while the supply side dimensions have been assessed, there is not much known or studied on the demand side barriers.

The study team would in due course address these for future and hopefully analyse information with some quantitative results.

6. PLOS authors have the option to publish the peer review history of their article (what does this mean? ). If published, this will include your full peer review and any attached files.

**Do you want your identity to be public for this peer review?** For information about this choice, including consent withdrawal, please see our Privacy Policy .

Reviewer #1: No

Reviewer #2: **Yes: ** Chase Emmanuel Egbo

Reviewer #3: **Yes: ** Lakshmi N Balaji
